# Evaluation of Exposure to Bisphenol Analogs through Canned and Ready-to-Eat Meal Consumption and Their Possible Effects on Blood Pressure and Heart Rate

**DOI:** 10.3390/nu16142275

**Published:** 2024-07-15

**Authors:** Merve Ekici, Nihan Çakır Biçer, Anıl Yirün, Göksun Demirel, Pınar Erkekoğlu

**Affiliations:** 1Department of Nutrition and Dietetics, Faculty of Health Sciences, Agri Ibrahim Cecen University, 04100 Agri, Turkey; mekici@agri.edu.tr; 2Department of Nutrition and Dietetics, Institute of Health Sciences, Acıbadem Mehmet Ali Aydınlar University, 34638 Istanbul, Turkey; 3Department of Nutrition and Dietetics, Faculty of Health Sciences, Acıbadem Mehmet Ali Aydınlar University, 34638 Istanbul, Turkey; 4Department of Toxicology, Faculty of Pharmacy, Cukurova University, 01250 Adana, Turkey; ayirun@cu.edu.tr (A.Y.); gdemirel@cu.edu.tr (G.D.); 5Department of Pharmaceutical Toxicology, Faculty of Pharmacy, Hacettepe University, 06430 Ankara, Turkey; erkekp@hacettepe.edu.tr; 6Department of Vaccine Technology, Vaccine Institute, Hacettepe University, 06430 Ankara, Turkey

**Keywords:** bisphenol A, bisphenol F, bisphenol S, blood pressure, canned food, ready-to-eat meal

## Abstract

Bisphenols are endocrine-disrupting chemicals used in plastics and resins for food packaging. This study aimed to evaluate the exposure to bisphenol A (BPA), bisphenol S (BPS), and bisphenol F (BPF) associated with the consumption of fresh, canned, and ready-to-eat meals and determine the effects of bisphenols on blood pressure and heart rate. Forty-eight healthy young adults were recruited for this study, and they were divided into the following three groups: fresh, canned, and ready-to-eat meal groups. Urine samples were collected 2, 4, and 6 h after meal consumption, and blood pressure and heart rate were measured. The consumption of ready-to-eat meals significantly increased urine BPA concentrations compared with canned and fresh meal consumption. No significant difference in BPS and BPF concentrations was observed between the groups. The consumption of ready-to-eat meals was associated with a significant increase in systolic blood pressure and pulse pressure and a marked decrease in diastolic blood pressure and heart rate. No significant differences were noted in blood pressure and heart rate with canned and fresh meal consumption. It can be concluded that total BPA concentration in consumed ready-to-eat meals is high. High BPA intake causes increase in urinary BPA concentrations, which may, in turn, lead to changes in some cardiovascular parameters.

## 1. Introduction

The modern lifestyle has significantly altered eating habits worldwide, leading to an increased demand for ready-to-eat meals in recent years. Owing to the increased frequency of consumption, ready-to-eat meals have become a significant dietary component for several individuals in today’s society, representing a significant pathway for potential exposure to various food contaminants [1].

Among the many contaminants in ready-made foods, bisphenol A (BPA), a widely used plasticizer in polycarbonate plastic and epoxy resin production, stands out as an abundant endocrine-disrupting chemical [2]. Polycarbonate is used in food items, including baby bottles, reusable plastic bottles, plates, cups, microwave ovenware, and storage containers, as well as in toys and pacifiers [3]. On the other hand, epoxy resins are commonly used as internal coatings in food and beverage cans to prevent direct contact between the food and metal, provide thermal stability, and enhance the mechanical strength of food-packaging jars [4]. Over the past few decades, owing to increased demand for BPA-containing products, the chemical has contributed to extensive environmental pollution, and it can contaminate food, water, and air. Humans are primarily exposed to BPA through diet, dermal contact, and inhalation [5].

Morgan et al. reported that 95% of BPA intake in the general population occurs through dietary sources, and dermal or respiratory exposures are relatively low [6]. Canned foods have been cited as the main dietary source of BPA intake [7]. BPA reaches its maximum concentration in the blood after 80 min following oral intake, and its half-life is 6 h. BPA is primarily conjugated with glucuronic acid and, to a lesser extent, with sulfate in the liver before being excreted via urine without entering enterohepatic circulation [8].

Recent epidemiological studies have identified significant associations between BPA exposure and chronic diseases, including diabetes, obesity, cardiovascular diseases, and renal dysfunction [3,9,10,11,12]. Considering these adverse effects, several countries have implemented measures, including banning BPA in baby bottles and food containers, establishing migration limits for BPA from food containers and beverage cans, and reducing the tolerable daily intake of BPA from 50 to 4 μg/kg body weight/day [13]. These restrictions, along with increased social awareness, have compelled the plastic industry to seek alternative chemicals to replace BPA [14].

Today, many polycarbonate plastics are labeled as BPA-free, due to the concerns of the general population. Although these products do not contain BPA in their composition, they typically maintain their structure and physical properties, probably as other bisphenol analogs are present in their chemical composition [15]. Currently, 16 bisphenol analogs are used in the chemical industry. Among these analogs, bisphenol S (BPS) and bisphenol F (BPF) are the main substitutes for BPA and are widely used in the production of inner linings for food containers, bottles, thermal papers, and food and beverage cans [16]. Additionally, animal experiments and epidemiological studies have identified significant associations between BPS and BPF exposure and oxidative stress, liver and thyroid function, obesity, and diabetes [17,18,19,20,21].

The prevalence of hypertension, the most effective risk factor for cardiovascular and renal diseases and the leading global burden of disease and mortality, is rapidly increasing worldwide. The World Health Organization has reported that approximately one billion individuals worldwide had hypertension in the year 2000, with this number expected to exceed approximately 1.5 billion by 2025 [14].

Several risk factors contribute to the development and pathogenesis of hypertension, among which environmental pollutants and chemicals play a significant role [22]. Environmental pollutants are a significant yet frequently overlooked risk factor for hypertension [14]. Some studies have suggested that air pollution, heavy metals, and other environmental pollutants are associated with the onset, progression, and severity of hypertension [23,24,25]. Reviewing the existing literature on the relationship between bisphenol derivatives and blood pressure (BP) reveals that many studies have cross-sectional designs. Moreover, they are usually conducted with non-standardized population groups and often measure only the effects of BPA exposure. Therefore, these studies have inconsistent results and do not directly measure the effects of other bisphenol analogs but only BPA [14,26,27,28].

This study aimed to assess exposure to BPA, BPF, and BPS after fresh, canned, and plastic-packaged ready-to-eat meal consumption and determine the effects of BPA, BPF, and BPS exposure on BP and heart rate (HR).

## 2. Materials and Methods

### 2.1. Study Population

In this randomized controlled single-blinded study conducted in May 2023, 48 healthy university students aged 18–30 years (24 females and 24 males) were recruited.

The inclusion criteria were being aged between 18 and 30 years, having a body mass index (BMI) between 18.5 and 24.9 kg/m^2^, absence of any diagnosed chronic illness, normotensive status, willingness to consume the provided meals as part of the study, and agreement to provide urine samples.

The exclusion criteria were having a known chronic illness or chronic nutritional disorder, eating disorder, or hypertension; use of medication, vitamins, or minerals; being pregnant or lactating; being in the menstrual period; and having a known history of acute bisphenol exposure. Researchers, except for the researcher providing the meals, were blinded, including those involved in BP measurement and urine sample analysis.

This study was approved by the Acıbadem Mehmet Ali Aydınlar University Medical Research Ethics Committee (ATADEK) (Ethics Committtee No: ATADEK-2023-2/42; Date: 27 January 2023) and was conducted in accordance with the Declaration of Helsinki. Written informed consent was obtained from all participants.

### 2.2. Dietary Intervention

Participants were asked to visit the study site twice within a four-day period. The dietary intervention was conducted at the Faculty of Health Sciences, Agri Ibrahim Cecen University. Participants were randomly allocated into the following three groups: fresh, canned, and plastic-packaged ready-to-eat meal groups, each comprising 16 individuals (8 males, 8 females). Before the intervention, participants were required to fill out a questionnaire to identify potential BPA sources.

On the first and second days, participants were instructed not to consume meals prepared outside the home, canned food and beverages, frozen meals, plastic-packaged food, water from polycarbonate bottles, food stored in plastic containers, or use plastic containers in microwave ovens. Furthermore, they were instructed not to consume any food or beverage (except water) after 10:00 p.m., to fast for at least 8 h, and to refrain from any physical activity beyond their daily routine. If opting for processed products, they were advised to select products in glass containers or, if not available, those in low-density polyethylene plastic containers (e.g., milk and orange juice). Moreover, they were asked not to consume coffee. However, participants who wanted to consume coffee were advised to use a French press or ceramic dripper instead of a plastic coffee machine. To evaluate whether the participants complied with the dietary rules specified in the study, 24 h food consumption records were taken, and their compliance with the protocol was checked. On the first and second days, they were instructed to collect their first urine sample in the morning after at least an 8 h overnight fasting.

On the third and fourth days, participants were instructed to visit the study site in the morning after at least an 8 h overnight fasting. At 10:00 a.m., participants assigned to random groups were instructed to consume the provided intervention meals. Following the intervention on the third day, all participants were instructed to adhere to the same guidelines as before the intervention. Until completion of the 6 h intervention procedure on the third and fourth days, they were asked not to consume any other food. Throughout the intervention, all participants remained in the laboratory setting. Additionally, they were allowed to engage in sedentary activities, including watching movies, reading books, playing games on electronic devices (computer and mobile phone), and other sedentary activities; however, they were not allowed to sleep. The algorithm of the diet intervention is shown in Figure 1.

### 2.3. Intervention Meals

Canned and plastic-packaged ready-to-eat meals were purchased from a local supermarket and stored together at +4 °C until the intervention day. While planning the intervention meals, care was taken to ensure that the same food in the supermarket had both canned and plastic-packaged ready-to-eat meal equivalents and that these were the most consumed products in Turkish food culture in order to ensure consumption by the participants. In addition, it was created by paying attention to the possibility of preparing the same meal as a fresh meal group, in accordance with canned and plastic-packaged ready-to-eat meals. It was ensured that the date and serial numbers of canned and plastic-packaged ready-to-eat meals were the same. Samples of each meal from every group were randomly selected and stored frozen at −20 °C for analysis. All meal samples were delivered to the Hacettepe University Faculty of Pharmacy Department of Pharmaceutical Toxicology Laboratory in a Styrofoam box with ice packs for analysis.

The working groups were determined as follows:(1)The canned meal group (*n* = 16): Participants consumed a meal comprising canned bean pilaki, canned stuffed grape leaves with olive oil and canned corn, canned chicken fillet, and a salad with canned tomatoes, along with 330 mL of glass-bottled water.(2)The plastic-packaged ready-to-eat meal group (*n* = 16): Participants consumed a meal comprising plastic-packaged ready-to-eat bean pilaki, plastic-packaged ready-to-eat stuffed grape leaves with olive oil, plastic-packaged ready-to-eat corn, plastic-packaged ready-to-eat chicken fillet, and a salad with plastic-packaged ready-to-eat tomatoes, with 330 mL of glass-bottled water.(3)The fresh meal group (*n* = 16): Similar to the other study groups, participants consumed a meal comprising bean pilaki, stuffed grape leaves with olive oil, corn, chicken fillet, and a salad with tomatoes, along with 330 mL of glass-bottled water.

The researcher prepared the foods in the fresh meal group. To prevent bisphenol migration, contact with plastic utensils and non-stick-coated cooking utensils was avoided, and porcelain, glass, wood, metal, and stainless steel kitchen utensils were used.

### 2.4. Urine Sample Collection

To remove all plastic materials, glass screw cap tubes were kept at 400 °C for 4 h. To remove any plastic residue, other glass materials were washed using n-hexane:tetrahydrofuran (1:1, *v*/*v*) for 4 h and subsequently dried in an incubator [29].

Subsequently, to prevent possible contamination from the screw caps, the containers were covered with aluminum foil. Participants were provided with pre-labeled deplasticized glass tubes for collecting urine samples. On the first and second days, participants were asked to collect their first morning urine and deliver these samples to the researcher in the morning. Moreover, on each intervention day, urine samples were collected from participants 2, 4, and 6 h after consuming the test meals.

Overall, eight urine samples were collected from each participant, and the samples were stored at −20 °C until the end of the intervention. At the end of the intervention, all urine samples were stored at −80 °C until the day of analysis. Two urine samples collected from each participant for each stage of the study (before and during intervention) were combined to form a single sample and analyzed.

### 2.5. BP and HR Measurements

The systolic BP (SBP), diastolic BP (DBP), and HR of the participants were measured using a sphygmomanometer (Omron M3 Comfort (HEM-7155-E(C) Kyoto, Japan) before and 2, 4, and 6 h after the diet intervention. Participants were instructed to remain in a seated position for at least 10 min, after which the first measurement was taken. After another 10 min, the second measurement was taken. The averages of two SBP and DBP measurements and two HR measurements were used for statistical analysis. Changes in BP and HR 2 h after meal consumption were used as the primary outcome measures. Pulse pressure (PP) was calculated as the difference between SBP and DBP. BP measurements were performed according to the guidelines of the American Heart Association [30].

### 2.6. Measurements of Bisphenol Analogs in Urine and Food

#### 2.6.1. Chemicals and Other Materials

All chemicals including bisphenol analogs, acetonitrile, n-hexane, and tetrahydrofuran were purchased from Sigma-Aldrich (Mannheim, Germany). Graphitized carbon column (GCB) was from Membrane Solutions, LLC (Nantong, China).

#### 2.6.2. Extraction of Bisphenol Derivatives from Urine Samples

To extract urine samples, the liquid–liquid extraction method was used. For this purpose, urine samples to be analyzed were centrifuged at 4000 rpm for 10 min, and 1000 μL of the urine sample was transferred to a 1.5 mL glass tube. Subsequently, 20 μL of β-glucuronidase/sulfatase enzyme combination was added to each urine sample, and the samples were incubated in a thermostatted water bath at 37 °C for 4 h. Next, samples were spiked with either 20 μL of BPA, BPF, or BPS (100 μg/L), vortexed vigorously for 30 s, and centrifuged at 12,000 rpm for 10 min. Following this process, the resulting supernatant was transferred to a 15 mL deplasticized glass tube and extracted with 4 mL ethyl acetate. The mixture was vortexed at 2500 rpm for 10 min and subsequently centrifuged at 4000 rpm for 10 min. The upper layer was transferred to a 10 mL glass tube and allowed to dry under gentle nitrogen flow. The residue was stored at −20 °C until analysis. BPA, BPF, and BPS analyses were separately performed. The limits of detection were 0.21, 0.88, and 0.84 ng/mL for BPA, BPF, and BPS, respectively. The limits of quantification were 0.54, 1.02, and 0.99 ng/mL for BPA, BPF, and BPS, respectively. After extraction, the samples stored at −20 °C were dissolved in 300 µL acetonitrile (60%, *v*/*v*, in water). Standards and samples were injected into high-performance liquid chromatography (HPLC) at 100 µL. The mobile phase comprised acetonitrile and 2.5% tetrahydrofuran (*v*/*v*, in water). The flow rate was 0.4 mL/min. Gradient elution was applied from 60:40 to 5:95 [31].

#### 2.6.3. Urinary Creatinine Determination

Urinary creatinine levels were measured for the normalization of the results. Urinary creatinine concentrations were analyzed using HPLC with slight modifications according to Jen et al. [32].

#### 2.6.4. Extraction of Bisphenol Derivatives from Canned and Ready-to-Eat Meals

The method used for the determination of bisphenol derivatives in food samples was adapted from the study by Cao et al. with modifications [33]. Food samples were homogenized using a stainless steel blender. The stainless steel stirrer was thoroughly cleaned (using detergents and solvents) after the homogenization of each food sample to prevent cross-contamination between samples and washed with n-hexane:tetrahydrofuran (1:1, *v*/*v*) and dried in an incubator for 4 h. Next, 1.0 g of the homogenized samples was weighed in a 15 mL deplasticized glass centrifuge tube, and after adding 5 mL of acetonitrile, the mixture was vortexed for 30 s. Subsequently, the tube was ultrasonicated for 30 min and centrifuged at 9000 rpm for 10 min. For the extraction step, the supernatant was transferred to a clean 50 mL glass centrifuge tube, diluted to 25 mL. Later, 50 µL of formic acid was added, and the mixture was vortexed for 60 sec. The mixture was then purified using a GCB column preconditioned with methanol (18 mL) and water (6 mL). Next, the GCB column was washed first with 6 mL of water and 6 mL of methanol/water (1:1, *v*/*v*) and, then, with 6 mL of methanol/acetone (4:1, *v*/*v*). The resulting eluent was dried under nitrogen stream. On the day of analysis, the residue was dissolved in 1 mL methanol/water (1:4, *v*/*v*) and applied to the HPLC. BPA, BPF, and BPS analyses were separately performed. The limits of detection for food samples were 0.52, 0.99, and 0.94 ng/mL for BPA, BPF, and BPS, respectively. The limits of quantification were 0.76, 1.14, and 1.07 ng/mL for BPA, BPF, and BPS, respectively.

### 2.7. Statistical Analyses

In summarizing the data obtained from the study, descriptive statistics including mean, standard deviation (SD), median (med), minimum (min), and maximum (max) values along with the significance level (*p*) are provided in tabular form for continuous (numerical) variables according to their distribution. Categorical variables are summarized as frequency (n) and percentage (%).

In the statistical analysis phase of the study, descriptive statistics for both dependent and independent variables are initially presented. Skewness and kurtosis values for variables are calculated within-group categories, and these values are within the range of (−2 ≤ skewness ≤ +2; −2 ≤ kurtosis ≤ +2), indicating compliance with normal distribution. Therefore, in this study, parametric tests were preferred. When examining relationships between categorical variables, Fisher’s test was applied in cases wherein at least one of the expected cell values was <5, and the Pearson chi-square independence test was used when all cell values were >5. To compare the means between two quantitative variables, the independent samples t-test was used. For comparisons of values measured at different times within groups (canned, ready-to-eat, and fresh meals), the repeated measures ANOVA test was employed. The independent samples ANOVA test was preferred for comparing measurement values between groups. In evaluating the results of the dependent samples ANOVA test, variance homogeneity was assessed using the Mauchly test. The sphericity test was used in cases wherein the homogeneity assumption was met; otherwise, the Greenhouse–Geisser test was applied. A significance level of α = 0.05 was set for all analyses, and statistical computations were performed using Statistical Package for the Social Sciences (SPSS, version 27, IBM, Armonk, NY, USA).

## 3. Results

The general characteristics of the participants are presented in Table 1. Each group comprised eight female and eight male participants. Almost all of the participants (*n* = 47, 97.9%) did not consume alcohol, and the proportion of the participants who smoked was also low (*p* > 0.05). The mean age of the participants was 21.75 ± 1.06, 22.31 ± 1.13, and 22.813 ± 0.834 years in the canned, ready-to-eat, and fresh meal groups, respectively. The canned meal group had a significantly lower mean age of participants than the fresh meal group (*p* < 0.05). All participants had normal BMI values.

The amounts of foods consumed and the concentrations of BPA, BPS, and BPF in the intervention meals, which were determined as standard for all participants in the study methodology, are presented in Table 2. In this study, the ready-to-eat meal group had total BPA, BPS, and BPF concentrations of 205.85, 5.91, and 10.16 ng/g, respectively, whereas the canned meal group had total BPA, BPS, and BPF concentrations of 127.03, 5.23, and 4.88 ng/g, respectively.

The intra and intergroup comparison results of urinary BPA, BPS, and BPF concentrations of the participants are presented in Table 3. Statistically significant differences were observed between the urinary BPA concentrations of participants in the canned, ready-to-eat, and fresh meal groups before and after consumption (*p* < 0.05). An examination of these differences showed that the mean urinary BPA concentration of participants in the ready-to-eat meal group at 0 and 2 h (12.615 ± 3.725 and 22.297 ± 6.368 μg/g creatinine, respectively) was significantly higher than those in the canned and fresh meal groups (9.149 ± 3.904, 16.323 ± 7.353 and 7.258 ± 1.314, 14.319 ± 3.268 μg/g creatinine, respectively) (*p* < 0.05). At the fourth hour, the mean urinary BPA concentration of participants in the ready-to-eat meal group (6.714 ± 2.420 μg/g creatinine) was significantly higher than those in the fresh meal group (3.968 ± 1.160 μg/g creatinine) (*p* < 0.05). Additionally, at the sixth hour, the mean urinary BPA concentration of participants in the canned and ready-to-eat meal groups (3.428 ± 1.462 and 3.451 ± 1.293 μg/g creatinine, respectively) was significantly higher than those in the fresh meal group (2.155 ± 0.712 μg/g creatinine) (*p* < 0.05). No statistically significant difference was noted between the urinary BPF and BPS concentrations among participants in the canned, ready-to-eat, and fresh meal groups (*p* > 0.05).

Upon intragroup comparisons of participants in the canned, ready-to-eat, and fresh meal groups, statistically significant differences were noted in the urinary BPA concentration at 0, 2, 4, and 6 h (*p* < 0.05). According to these findings, the urinary BPA concentration in all the three groups increased at the second hour and significantly decreased at the fourth and sixth hours (urinary BPA at 6th hour < 4th hour < 0th hour < 2nd hour).

Upon intragroup comparisons of participants in the canned meal group, no statistically significant differences were observed in the urinary BPS and BPF concentrations at 0, 2, 4, and 6 h (*p* > 0.05). However, upon intragroup comparisons of participants in the ready-to-eat and fresh meal groups, statistically significant differences were observed in the urinary BPS and BPF concentrations at 0, 2, 4, and 6 h (*p* < 0.05). Accordingly, in the ready-to-eat meal and fresh meal groups, the urinary BPS concentration increased at the second hour and significantly decreased between the fourth and sixth hours (urinary BPS at 6th hour < 4th hour < 0th hour < 2nd hour). For the participants in the ready-to-eat meal group, the urinary BPF concentration increased at the second hour and significantly decreased between the fourth and sixth hours (urinary BPF at 6th hour < 4th hour < 0th hour < 2nd hour). Participants in the fresh meal group showed a significant decrease in BPF concentration in urine samples at 4 h compared with that at 2 h (urinary BPF at 4th hour < 2nd hour).

The changes in SBP, DBP, HR, and PP of participants before and after the consumption of meals are illustrated in Table 4. Intergroup comparisons showed no statistically significant difference in HR and PP values of participants in the canned, ready-to-eat, and fresh meal groups at 0, 2, 4, and 6 h (*p* > 0.05). Similarly, no significant difference was observed in SBP and DBP values at 0, 2, and 4 h (*p* > 0.05). However, at the sixth hour, a statistically significant difference was identified in SBP and DBP values among participants in the canned, ready-to-eat, and fresh meal groups (*p* < 0.05). Upon examining this difference, participants in the canned meal group showed significantly higher SBP and DBP values than those in the ready-to-eat meal group.

Intragroup comparisons of participants in the canned meal group showed no statistically significant difference in SBP and PP values at 0, 2, 4, and 6 h (*p* > 0.05). However, statistically significant differences were detected in DBP and HR values at 0, 2, 4, and 6 h (*p* < 0.05). Upon analyzing these differences, participants in the canned meal group showed significantly lower DBP and HR values at the 4th hour than those at the 6th hour.

Participants in the ready-to-eat meal group exhibited statistically significant differences in SBP, DBP, HR, and PP values at 0, 2, 4, and 6 h (*p* < 0.05). Examination of these differences revealed that the SBP and HR values of participants in the ready-to-eat meal group were significantly higher at the second hour. Additionally, participants in the ready-to-eat meal group showed a significant decrease in DBP and PP values at 0, 2, 4, and 6 h. In the fresh meal group, intragroup comparison results showed no statistically significant difference in SBP, DBP, HR, and PP values at 0, 2, 4, and 6 h (*p* > 0.05).

## 4. Discussion

The consumption of ready-to-eat and canned meals has been recently increasing due to changes in lifestyle. The materials used in the packaging of such foods may have various effects on health. To our knowledge, this study is the first to evaluate the effects of BPA, BPS, and BPF consumption on BP and HR in healthy young adults through interventions involving fresh, canned, and ready-to-eat meal groups.

The primary source of human exposure to bisphenols is through packaged foods and beverages [34]. Intervention studies examining the contribution of food packaging to bisphenol exposure typically include canned and fresh food groups and frequently focus solely on BPA exposure [27,35,36,37]. However, owing to bans on BPA use, BPA analogs such as BPS and BPF are widely used in the plastic industry [38]. Lifestyle changes have led to an increased demand for ready-to-eat meals packaged in plastic, alongside canned foods [39]. The higher level of industrial processing typically associated with ready-to-eat meals (storage, cooking, packaging, and refrigeration) can significantly increase the risk of bisphenol contamination [1]. Furthermore, factors including storage duration and temperature, heat applied for sterilization, contact surface, type of plastic material, type of food, and packaging temperatures contribute to the migration of bisphenols from packaging materials to food [40].

Upon examining the contribution of food packaging to bisphenol exposure, consuming ready-to-eat meals significantly increased urinary BPA concentrations compared with consuming the same meal as canned or fresh. Although the highest increase in urinary BPS and BPF concentrations was observed with ready-to-eat meal consumption, it was not statistically significant across the groups. This situation is attributed to ready-to-eat meals containing higher BPA, BPS, and BPF concentrations than meals in the canned and fresh meal groups. The ready-to-eat meal group had higher total BPA, BPS, and BPF concentrations than the canned meal group. Owing to the migration of bisphenols from packaging materials to food and beverages, the smell, color, and taste characteristics of products change, potentially leading to various harmful effects on human health [38].

Before the intervention, during a period when multiple dietary sources of bisphenol exposure were controlled, urinary BPA concentrations at 0 h were high in all the three groups. These common exposures are likely associated with direct contact with BPA-containing materials or BPA exposure in household dust or indoor air [35].

In all the three groups, urinary BPA, BPS, and BPF concentrations peaked at the second hour and subsequently decreased. Similarly, in a study by Teeguarden et al., the time to reach peak urinary BPA concentrations was 2.75 h [41], whereas in the study by Peng et al., urinary BPA concentrations peaked at the fourth hour [37]. BPA has a half-life of approximately 6 h, and the elimination of orally administered BPA is completed within 24 h [42].

Statistically significant differences were observed in urinary BPA concentrations among participants in the canned, ready-to-eat, and fresh meal groups following meal consumption. An examination of these differences revealed that the ready-to-eat meal group had significantly higher urinary BPA concentrations than the canned and fresh meal groups at the 2nd hour. Furthermore, at the 4th hour, the urinary BPA concentrations of participants in the ready-to-eat meal group were significantly higher than those in the fresh meal group. Moreover, at the 6th hour, participants in the canned and ready-to-eat meal groups exhibited markedly higher urinary BPA concentrations than those in the fresh meal group. These findings indicate that consuming ready-to-eat meals significantly increases urinary BPA concentrations compared with consuming the same meal as canned or fresh. This situation is attributed to ready-to-eat meals having higher BPA concentrations.

No significant differences were noted in urinary BPS and BPF concentrations among participants in the canned, ready-to-eat, and fresh meal groups following meal consumption. However, the greatest increase in urinary BPS and BPF concentrations in the second hour after meal consumption was observed in participants in the ready-to-eat meal group. This result is likely because ready-to-eat meals have higher BPS and BPF concentrations.

In this study, the 2nd-hour urine BPA concentrations of participants in the canned, ready-to-eat, and fresh meal groups showed a significant decrease between hours. However, urine BPS concentrations at the 2nd hour significantly decreased between hours in the ready-to-eat and fresh meal groups, whereas no significant decrease was observed in the canned meal group. Similarly, urine BPF concentrations at the 2nd hour showed a significant decrease between hours only in participants in the ready-to-eat meal group. These findings indicate that the magnitude of change in urinary BPA, BPS, and BPF concentrations varies with the consumption of canned, ready-to-eat, and fresh meals, and their change profiles differ over time.

Studies have demonstrated that consuming canned meals significantly increases urinary BPA concentrations compared with consuming fresh foods [28,35,37]. In the present study, consuming canned meals increased urinary BPA concentrations compared with consuming the same meal as a fresh meal; however, the difference was only significant at the 6th hour. Additionally, increases in urinary BPA, BPS, and BPF concentrations were observed in the 2nd hour following the consumption of meals in the fresh meal group. Previous studies have shown no significant increase in urinary BPA concentrations when consuming unpackaged fresh foods, such as fresh fish and vegetables [28,35,37]. However, in this study, packaged foods purchased from the market (rice, beans, chicken, and oil) were used to prepare the fresh meal group as counterparts to canned and ready-to-eat meals. Therefore, it is believed that the meals in the fresh meal group may have been contaminated with these bisphenols, and it is presumed that this contamination occurred during the transportation, storage, or production stages of the food’s raw materials. Consequently, it is believed that the increase in urinary BPA, BPS, and BPF concentrations in the fresh meal group is because of this contamination.

In this study, the acute effects of BPA, BPS, and BPF exposure on BP, PP, and HR were also determined. In the fresh meal group, no significant relationship was noted between BPA, BPS, and BPF exposure and BP, PP, and HR. At the 2nd hour following canned and ready-to-eat meal consumption, an increase in SBP and PP was observed, whereas a decrease in DBP and HR was detected. However, this was significant only in participants of the ready-to-eat meal group. Additionally, when examining the urinary bisphenol concentrations at the 2nd hour, participants in the ready-to-eat meal group had significantly higher urinary BPA concentrations, with the most pronounced increase observed in urinary BPS and BPF concentrations among participants in the ready-to-eat meal group. These results demonstrate the acute effects of increased BPA, BPS, and BPF exposure due to ready-to-eat meal consumption on SBP and HR. Moreover, participants in the ready-to-eat meal group showed a significant decrease in SBP and PP at the 4th hour, whereas the decrease in DBP and HR values significantly continued at the 4th and 6th hours. The increase in PP may be a consequence of the increase in SBP and the decrease in DBP. Furthermore, a decrease in PP is generally indicative of reduced cardiac function [43]. An in vitro study showed that BPA could increase the activity of high-conductance Ca^2+^/voltage-sensitive K^+^ channels by binding to estrogenic receptors in coronary arterial smooth muscle cells, thereby affecting heart function [44]. Therefore, the decrease in HR could be one of the estrogenic effects of BPA and its analogs [45]. Similarly, Bae et al. reported that BPA exposure reduced HR [26]. However, the same authors later concluded in another study that BPA exposure did not affect HR [27].

Participants in the canned meal group had significantly higher SBP and DBP values at the 6th hour than those in the ready-to-eat meal group. Furthermore, although participants in the canned meal group showed a significant increase in SBP, DBP, and HR values at the 6th hour compared with those at the 4th hour, this increase was significant only for DBP and HR values. Additionally, when examining urinary bisphenol concentrations at the 6th hour, participants in the canned and ready-to-eat meal groups had significantly higher urinary BPA concentrations than those in the fresh meal group, with no significant difference observed in urinary BPS and BPF concentrations. Although participants in the canned meal group showed a significant decrease in urinary BPA concentrations during the intervention period, no significant decrease was observed in urinary BPS and BPF concentrations. Conversely, during the intervention period, participants in the ready-to-eat meal group showed a significant decrease in urinary BPA, BPS, and BPF concentrations. This result suggests that the impact of BPA, BPS, and BPF exposure on blood pressure decreased in participants in the ready-to-eat meal group at the 6th hour. Therefore, the significant increase in SBP and DBP values at the 6th hour in the canned meal group compared with that in the ready-to-eat meal group may be attributed to the synergistic effect of BPA, BPS, and BPF exposure. A previous study also suggested that simultaneous exposure to bisphenol analogs along with low BPA concentrations lead to more severe cardiovascular or cardiometabolic disorders or toxicity than exposure to high BPA concentrations alone because of the synergistic effect [46].

This study had some limitations. First, in the preparation of meals in the fresh meal group, packaged foods purchased from the market (rice, beans, chicken, and oil) were used. Therefore, the contamination of the fresh meals with bisphenol analogs could not be controlled, and as a result, the consumption of fresh meals led to an increase in urinary BPA, BPS, and BPF concentrations. Second, BPA, BPS, and BPF concentrations in the meals of the fresh meal group were not measured. Third, we did not consider the possibility of other chemicals that may have migrated from the canned and plastic packaging to the foods. Therefore, in the BP analysis, we did not account for the potential confounding effects of other chemicals. However, other chemicals present in canned and ready-to-eat meals including residue from unknown factors in BP measurements could also affect the results.

## 5. Conclusions

This study demonstrates that consuming ready-to-eat meals increases BPA, BPS, and BPF exposure compared with consuming canned and fresh meals. Furthermore, increased BPA, BPS, and BPF exposure due to the consumption of ready-to-eat meals leads to an increase in BP and induces changes in some cardiovascular parameters. This result suggests that increased bisphenol exposure through plastic-packaged ready-to-eat meal consumption potentially elevates health risks at a societal level. The findings of this study underscore the necessity for further research to understand the potential relationship between bisphenol exposure from plastic-packaged ready-to-eat meals and cardiovascular health.

## Figures and Tables

**Figure 1 nutrients-16-02275-f001:**
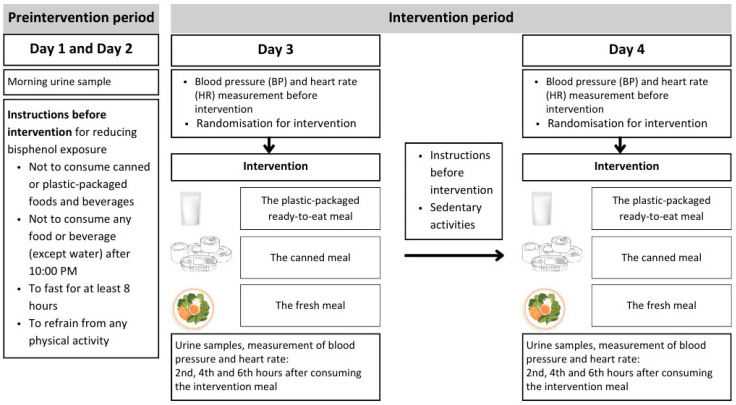
The algorithm of the diet intervention.

**Table 1 nutrients-16-02275-t001:** General characteristics of the participants.

	Canned Meal Group ^a^ (*n* = 16)		Ready-to-Eat Meal Group ^b^ (*n* = 16)		Fresh Meal Group ^c^ (*n* = 16)		Test Statistic	*p*
	*n*	%	*n*	%	*n*	%		
Sex								
Male	8	50.00	8	50.00	8	50.00	<0.001	1.000
Female	8	50.00	8	50.00	8	50.00		
Marital status								
Single	16	100.00	16	100.00	16	100.00	-	-
Alcohol use								
No	16	100.00	15	93.75	16	100.00	1.859	1.000
Yes	0	0.00	1	6.25	0	0.00		
Smoking								
No	10	62.50	14	87.50	11	68.75	2.778	0.254
Yes	6	37.50	2	12.50	5	32.25		
Age (year) (Mean ± SD)	21.75 ± 1.06		22.313 ± 1.138		22.813 ± 0.834		4.340	0.019 * (a and c)
BMI (kg/m^2^) (Mean ± SD)	21.753 ± 2.211		21.733 ± 2.021		21.982 ± 2.242		0.066	0.936

a, canned meal group; b, ready-to-eat meal group; c, fresh meal group; *n*, number; BMI, body mass index; SD: standard deviation * *p* < 0.05.

**Table 2 nutrients-16-02275-t002:** The consumed amounts and BPA, BPS, and BPF concentrations of the meals.

	Consumed Amount (g)	Canned Meal (ng/g)			Plastic-Packaged Ready-to-Eat Meal (ng/g)		
		BPA	BPS	BPF	BPA	BPS	BPF
Bean pilaki	200	55.47	1.48	2.33	56.23	1.02	1.13
Stuffed grape leaves with olive oil	100	64.12	2.54	1.54	145.97	3.86	7.89
Corn	100	1.54	-	-	1.18	-	-
Chicken fillet	100	4.76	1.21	1.01	1.45	1.03	1.14
Tomatoes	100	1.14	-	-	1.02	-	-
Total		127.03	5.23	4.88	205.85	5.91	10.16

BPA, bisphenol A; BPS, bisphenol S; BPF, bisphenol F.

**Table 3 nutrients-16-02275-t003:** Urinary BPA, BPS, and BPF concentrations of the meal groups.

μg/g Creatinine	Canned Meal Group ^a^ (*n* = 16)					Ready-to-Eat Meal Group ^b^ (*n* = 16)					Fresh Meal Group ^c^ (*n* = 16)					Test Statistic	*p*
	Mean	SD	Med	Min	Max	Mean	SD	Med	Min	Max	Mean	SD	Med	Min	Max		
0th hour BPA ^I^	9.149	3.904	8.670	4.260	17.540	12.615	3.725	13.270	7.510	18.740	7.258	1.314	7.315	5.460	9.870	11.485	<0.001 * (b > a, c)
2nd hour BPA ^II^	16.323	7.353	15.330	7.520	31.570	22.297	6.368	22.965	14.010	31.590	14.319	3.268	14.250	9.960	20.740	7.851	0.001 * (b > a, c)
4th hour BPA ^III^	5.469	1.860	5.945	2.030	8.570	6.714	2.420	6.110	3.410	12.370	3.968	1.160	3.935	2.180	6.020	8.510	0.001 * (b > c)
6th hour BPA ^IV^	3.428	1.462	3.460	1.170	6.510	3.451	1.293	3.140	1.510	6.210	2.155	0.712	2.185	1.190	4.010	6.116	0.004 * (a, b > c)
Intragroup comparison	F = 50.574 *p* < 0.001 * (IV < III < I < II)					F = 168.760 *p* < 0.001 * (IV < III < I < II)					F = 268.963 *p* < 0.001 * (IV < III < I < II)						
0th hour BPS ^I^	2.666	0.876	2.710	1.210	3.650	2.563	1.032	2.435	1.020	4.210	3.098	0.636	3.100	2.320	3.870	0.487	0.622
2nd hour BPS ^II^	2.839	1.661	2.650	1.060	5.740	3.508	1.996	4.025	1.020	6.910	3.566	2.465	3.270	1.130	6.960	0.400	0.674
4th hour BPS ^III^	1.480	0.560	1.220	1.020	2.560	1.829	0.577	1.990	1.080	2.650	2.000	0.696	2.220	1.020	2.540	1.234	0.316
6th hour BPS ^IV^	1.315	0.464	1.100	1.050	2.010	1.135	0.089	1.145	1.020	1.230	1.197	0.112	1.240	1.070	1.280	0.38	0.694
Intragroup comparison	F = 79.168 *p* = 0.071					F = 118.114 *p* < 0.001 * (IV < III < I < II)					F = 268.963 *p* < 0.001 * (IV < III < I < II)						
0th hour BPF ^I^	2.494	0.998	2.205	1.300	4.520	3.181	0.895	3.170	2.140	4.780	2.297	0.760	2.310	1.020	3.180	2.764	0.080
2nd hour BPF ^II^	3.188	1.745	3.405	1.070	7.740	4.171	2.372	4.330	1.050	8.210	3.059	1.910	2.990	1.020	7.010	1.405	0.256
4th hour BPF ^III^	1.513	0.631	1.280	1.010	3.070	1.929	0.646	2.120	1.020	2.990	1.499	0.586	1.170	1.030	2.560	1.549	0.231
6th hour BPF ^IV^	1.086	0.065	1.120	1.010	1.140	1.154	0.129	1.170	1.010	1.340	1.130	0.113	1.130	1.050	1.210	0.577	0.578
Intragroup comparison	F = 9.331 *p* = 0.092					F = 74.293 *p* < 0.001 * (IV < III < I < II)					F = 192.905 *p* = 0.046 * (III < II)						

a, canned meal group; b, ready-to-eat meal group; c, fresh meal group; I, 0th hour; II, 2nd hour; III, 4th hour; IV, 6th hour; BPA, bisphenol A; BPS, bisphenol S; BPF, bisphenol F; SD, standard deviation; Med, median; Min, minimum; Max, maximum; *p*, significance level; F, repeated measures ANOVA test. * *p* < 0.05.

**Table 4 nutrients-16-02275-t004:** SBP, DBP, HR, and PP measurements of the meal groups.

	Canned Meal Group ^a^ (*n* = 16)					Ready-to-Eat Meal Group ^b^ (*n* = 16)					Fresh Meal Group ^c^ (*n* = 16)					Test Statistic	*p*
	Mean	SD	Med	Min	Max	Mean	SD	Med	Min	Max	Mean	SD	Med	Min	Max		
0th hour SBP ^I^ (mmHg)	106.969	9.231	106.75	93.00	122.50	104.313	10.274	101.50	92.00	127.50	102.625	7.564	105.25	86.00	111.50	0.928	0.403
2nd hour SBP ^II^ (mmHg)	107.719	6.666	108.50	97.50	119.50	106.719	8.989	102.50	97.00	124.50	101.781	9.690	100.50	84.50	117.00	2.214	0.121
4th hour SBP ^III^ (mmHg)	103.781	10.116	102.00	88.00	121.50	101.688	8.625	101.25	83.50	114.50	102.156	10.759	103.00	79.00	117.00	0.198	0.821
6th hour SBP ^IV^ (mmHg)	109.156	10.473	107.25	92.50	128.00	99.094	6.741	98.25	87.00	112.00	102.313	8.735	101.75	88.50	122.00	5.477	0.007 * (a > b)
Intragroup comparison	F = 1.645 *p* = 0.192					F = 5.007 *p* = 0.004 * (IV < III < I < II)					F = 0.071 *p* = 0.975						
0th hour DBP ^I^ (mmHg)	76.125	9.375	77.25	59.50	91.50	76.250	9.918	76.00	64.50	103.50	70.625	7.336	69.00	61.00	88.50	2.063	0.139
2nd hour DBP ^II^ (mmHg)	74.469	8.316	76.50	61.00	91.00	73.406	8.862	73.25	59.50	97.00	69.469	8.908	69.00	51.00	82.00	1.467	0.241
4th hour DBP ^III^ (mmHg)	71.094	7.515	69.25	58.00	91.00	72.281	5.814	72.75	63.00	81.50	71.031	6.702	71.50	55.50	80.00	0.176	0.839
6th hour DBP ^IV^ (mmHg)	77.094	10.364	79.25	60.50	94.50	68.438	8.652	67.50	50.50	85.50	69.344	8.189	68.50	56.00	86.00	4.358	0.019 * (a > b)
Intragroup comparison	F = 4.968 *p* = 0.005 * (III < IV)					F = 5.687 *p* = 0.002 * (IV < III < II < I)					F = 0.484 *p* = 0.695						
0th hour PP ^I^ (mmHg)	30.844	8.080	30.50	15.00	43.50	28.063	8.296	25.75	18.50	50.00	32.000	9.680	32.00	14.50	47.00	0.863	0.429
2nd hour PP ^II^ (mmHg)	33.250	7.045	32.75	23.50	44.50	33.313	12.468	29.25	18.50	57.50	32.313	10.896	32.50	14.50	49.50	0.047	0.955
4th hour PP ^III^ (mmHg)	32.688	12.015	35.00	0.00	51.00	29.406	7.638	30.25	19.00	40.50	31.125	9.821	31.75	13.00	47.00	0.432	0.652
6th hour PP ^IV^ (mmHg)	32.063	12.025	31.50	0.00	54.00	30.656	6.755	28.00	25.00	45.50	32.969	8.812	33.50	21.50	47.00	0.243	0.785
Intragroup comparison	F = 0.339 *p* = 0.798					F = 3.095 *p* = 0.036 * (I < III < IV < II)					F = 0.283 *p* = 0.837						
0th hour HR ^I^ (bpm)	85.688	12.389	85.50	67.00	106.00	88.125	12.425	86.25	74.00	125.50	80.625	13.316	77.00	57.00	112.00	1.448	0.246
2nd hour HR ^II^ (bpm)	82.969	8.904	83.25	64.50	95.00	85.094	11.578	84.00	71.00	119.00	76.219	12.518	80.75	54.50	99.00	2.786	0.072
4th hour HR ^III^ (bpm)	78.438	8.491	78.25	66.50	93.00	82.938	12.480	79.25	67.50	117.00	77.781	11.207	80.50	61.00	94.50	1.070	0.352
6th hour HR ^IV^ (bpm)	82.500	11.182	82.25	64.50	105.50	81.844	10.281	80.00	69.50	108.50	77.281	13.953	80.00	52.50	95.00	0.912	0.409
Intragroup comparison	F = 3.211 *p* = 0.032 * (III < IV)					F = 3.207 *p* = 0.032 * (a > b > c > d)					F = 1.621 *p* = 0.198						

a, canned meal group; b, ready-to-eat meal group; c, fresh meal group; I, 0th hour; II, 2nd hour; III, 4th hour; IV, 6th hour; SBP, systolic blood pressure; DBP, diastolic blood pressure; PP, pulse pressure; HR, heart rate; SD, standard deviation; Med, median; Min, minimum; Max, maximum, *p*, significance level; F, repeated measures ANOVA test. * *p* < 0.054.

## Data Availability

The data that support the findings of this study are available from the corresponding author, [N.Ç.B.], upon reasonable request.

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
