# Peer review of "Evaluation of Exposure to Bisphenol Analogs through Canned and Ready-to-Eat Meal Consumption and Their Possible Effects on Blood Pressure and Heart Rate"

_nutrients, 2024, doi:10.3390/nu16142275_

Round 1

Reviewer 1 Report

Comments and Suggestions for Authors

1. Comment

In the Abstract (lines 24 to 25), the authors state, “Consumption of ready-to-eat meals significantly increased urine BPA concentrations compared with canned and fresh meal consumption.” However, according to Table 3, all three groups showed a statistically significant increase in urine BPA concentrations compared to the baseline (i.e., 0th hour value). Therefore, this statement appears misleading.

2. Comment

In the Abstract (lines 29 to 30), the authors state, “It can be concluded that BPA concentration in ready-to-eat meals is high.” Could the authors clarify where the evidence in the manuscript supports this conclusion? If the evidence comes from Table 2, it raises the question of whether the total BPA values in canned meals versus plastic-packaged ready-to-eat meals could be similar or even in the opposite direction simply because of the selection of food items in this study. For example, the BPA concentrations from corn, chicken fillet, and tomatoes are numerically lower in ready-to-eat meals than in canned meals. If the studied meal excluded Stuffed grape leaves with olive oil, could the authors still reach such a conclusion?

3. Comment

In Section 2.2 (lines 127 and 130), the authors state that the subjects could opt for processed products and coffee, and these could be considered study protocol deviations. Did the authors keep a record of these protocol deviations? Could the authors discuss whether and to what extent these deviations affected the study and its conclusions? 

4. Comment

In lines 288 to 289, the authors state, “The consumed amounts and BPA, BPS, and BPF concentrations… are presented in Table 2.” The exact meaning of “consumed amounts” is unclear. For example, does this refer to the amount of food provided to each subject (where subjects could choose to eat all or part of it), or does it refer to the actual amount each subject ate?

5. Comment

Questions about Table 3.

First, the baseline BPA concentrations in the ready-to-eat meal group were statistically significantly higher than those in the canned meal group and fresh meal group. This raises the question of how comparable the three groups truly are at the baseline level. Additionally, did the authors adjust for this baseline factor when reaching their conclusion? Could the outcomes (e.g., blood pressure and heart rate) observed in this study be affected by multiple factors, including the high baseline BPA concentrations in the ready-to-eat meal group?

Second, 0th hour BPA, BPS, and BPF samples were collected from Day 1 and Day 2, which are considered “bisphenol washout days” intended to reduce chemical exposure. Could the authors explain why, even after 2 days of this activity, are the baseline values still higher than the 6th hour values after bisphenol intake (and such a difference is statistically significant)?

Third, did the authors compare the metabolism rate of BPA among three different groups? For example, by comparing the concentration curve (time after BPA intake versus BPA concentration) to investigate if these three groups had similar metabolism rate of BPA. In other words, differences in BPA metabolism among the three groups could potentially contribute to the outcomes observed in the study (e.g., blood pressure and heart rate).

6. Comment

Discrepancy between statement and data.

First, in lines 365 to 366, the authors state, “Intragroup comparisons of participants in the canned meal group showed no statistically significant difference in SBP and HR values at 0, 2, 4, and 6 h (p > 0.05).” However, according to Table 4, there is statistically significant difference in HR (p = 0.032).

Second, in lines 366 to 367, the authors state, “However, statistically significant differences were detected in DBP and PP values at 0, 2, 4, and 6 h (p < 0.05).” However, according to Table 4, there is no statistically significant difference in PP (p = 0.789).

Third, in lines 368 to 369, the authors state, “Upon analyzing these differences, participants in the canned meal group showed significantly lower DBP and PP values at the 4th hour than those at the 6th hour.” However, according to Table 4, the PP value at 6th hour is not statistically different from that at 4th hour.

7. Comment

Typographical errors.

First, in line 236, “and washed and washed”.

Second, in line 217, the unit “ng/mL” should come after 0.84. In line 248, the unit “ng/mL” should come after 0.94.

Third, in the footnote of Table 1, the “*” is missing from “p < 0.05”.

Forth, in both Tables 3 and 4, “a, b, and c” were used to indicate “canned meal group, ready-to-eat group, and fresh meal group”. Meanwhile, “a, b, c, and d” were also used to indicate “0, 2, 4, 6 hour”. This could potentially cause confusion for the audience. The authors might consider using different symbols (e.g., I, II, III, IV) to represent the hours instead.

Author Response

Response to Reviewer 1

Thank you very much for taking the time to review this manuscript. Please find the detailed responses below and the corresponding revisions/corrections highlighted changes in the re-submitted files.

Point 1: In the Abstract (lines 24 to 25), the authors state, “Consumption of ready-to-eat meals significantly increased urine BPA concentrations compared with canned and fresh meal consumption.” However, according to Table 3, all three groups showed a statistically significant increase in urine BPA concentrations compared to the baseline (i.e., 0th hour value). Therefore, this statement appears misleading.

Response 1: In the Abstract (lines 24 to 25), the sentence "Consumption of ready-to-eat meals significantly increased urine BPA concentrations compared with canned and fresh meal consumption." refers to the result of the comparison of the urine BPA concentrations of the ready-to-eat, canned and fresh meal groups at 2 hours between the groups. In Table 3, the comparison result of urinary BPA concentrations at the 2nd hour is shown as p= 0.001* (b > a, c), and "a, b and c" are used to indicate "canned food group, ready-to-eat food group, and fresh food group" respectively. "a, b, c, and d" in the first column are used to indicate "0th, 2nd, 4th, 6th hour" respectively. In the first column "a, b, c and d" are used to indicate "0th, 2nd, 4th, 6th hour" respectively. This was considered misleading and was corrected using different symbols (e.g. I, II, III, IV) to represent the hours in the first column.

Point 2: In the Abstract (lines 29 to 30), the authors state, “It can be concluded that BPA concentration in ready-to-eat meals is high.” Could the authors clarify where the evidence in the manuscript supports this conclusion? If the evidence comes from Table 2, it raises the question of whether the total BPA values in canned meals versus plastic-packaged ready-to-eat meals could be similar or even in the opposite direction simply because of the selection of food items in this study. For example, the BPA concentrations from corn, chicken fillet, and tomatoes are numerically lower in ready-to-eat meals than in canned meals. If the studied meal excluded Stuffed grape leaves with olive oil, could the authors still reach such a conclusion?

Response 2: In the Abstract (lines 29 to 30), the sentence “It can be concluded that BPA concentration in ready-to-eat meals is high.” has been corrected to “It can be concluded that the total BPA concentration in consumed ready-to-eat meals is high.”

Point 3: In Section 2.2 (lines 127 and 130), the authors state that the subjects could opt for processed products and coffee, and these could be considered study protocol deviations. Did the authors keep a record of these protocol deviations? Could the authors discuss whether and to what extent these deviations affected the study and its conclusions? 

Response 3:

To evaluate whether the participants complied with the rules specified in the study, particularly in the “washout period”, 24-hour food consumption records were taken during the study period. According to food consumption records, participants complied with the study protocol outlined in Section 2.2 (lines 121-134). According to food consumption records, the participants did not consume the packaged products mentioned in lines 128 and 132. It is stated by adding the explanation as “To evaluate whether the participants complied with the dietary rules specified in the study, 24-hour food consumption records were taken and their compliance with the protocol was checked.” (lines 132-134). Additionally, the amount of BPA they were exposed to through diet was calculated from food consumption records. The average values ​​specified by the European Food Safety Authority (EFSA) are taken as the basis for the amount of BPA contained in foods. There was no statistically significant difference between BPA exposure levels based on food groups calculated according to the food consumption record for participants in the ready food, canned food, and fresh food groups (p>0.05).

Point 4: In lines 288 to 289, the authors state, “The consumed amounts and BPA, BPS, and BPF concentrations… are presented in Table 2.” The exact meaning of “consumed amounts” is unclear. For example, does this refer to the amount of food provided to each subject (where subjects could choose to eat all or part of it), or does it refer to the actual amount each subject ate?

Response 4: The expression "consumed amounts" refers to the amount of food given to all participants as standardized by the study methodology and consumed by the participants. In lines 291-293, the sentence is changed as “The amounts of foods consumed and the concentrations of BPA, BPS, and BPF in the intervention meals, which were determined as standard for all participants in the study methodology, are presented in Table 2.” In section 2.1 (line 104) it is also stated as an inclusion criterion as "willingness to consume the provided meals as part of the study". Therefore, each participant consumed all of the provided quantities.

Point 5: Questions about Table 3.

Point 5, Question 1: First, the baseline BPA concentrations in the ready-to-eat meal group were statistically significantly higher than those in the canned meal group and fresh meal group. This raises the question of how comparable the three groups truly are at the baseline level. Additionally, did the authors adjust for this baseline factor when reaching their conclusion? Could the outcomes (e.g., blood pressure and heart rate) observed in this study be affected by multiple factors, including the high baseline BPA concentrations in the ready-to-eat meal group?

Response 5, Question 1: Due to widespread use, humans are continuously exposed to BPA by diet (oral), dermal, and inhalation routes. Considering the highest exposure is through the oral route, dietary exposure to BPA and its analogues has been investigated in this descriptive study. It was thought that there should be some time to lower the dietary exposure to BPA and that is why a period of time was needed in order to decrease BPA in the body and to later investigate the effects of diet on the concentration of BPA in biological fluids and food consumption records for these days were also taken and checked for deviations. By calculating the dietary exposure of BPA from food consumption records, it was determined that there was no statistically significant difference between the groups. Therefore, we decided that the groups were comparable since there was no difference in dietary exposure between the groups at hour 0. We can suggest that urinary BPA concentration values ​​at the hour 0 are thought to originate from dermal absorption (personal care products such as thermal paper, paint, shampoo, shower gel, dental materials, and cosmetics) and inhalation, which are other exposure sources of BPA. In order to further explain this issue, we have explained this in the first submitted manuscript as “Before the intervention, during a period when multiple dietary sources of bisphenol exposure were controlled, urinary BPA concentrations at 0 h were high in all the three groups. These common exposures are likely associated with direct contact with BPA containing materials or BPA exposure in household dust or indoor air [35].” (lines 417-420).

Although the urinary BPA concentration at the hour 0 was significantly higher in the ready meal group than in the canned and fresh food groups, there was no statistically significant difference in the SBP, DBP, PP and HR values ​​at the zero hour between the groups (p>0.05). In addition, it was determined that the DBP values ​​at 0th hour in the ready meal and canned food groups (76.250 ± 9.918 mmHg, 76.125 ± 9.375 mmHg respectively) were similar. SBP and PP values ​​at the zero hour were found to be higher in the canned food group (106.969 ± 9.231 mmHg, 30.844 ± 8.080 mmHg respectively) than in the ready meal group (104.313 ± 10.274 mmHg, 28.063 ± 8.296 mmHg respectively). Therefore, if the urinary BPA concentration at the zero hour had affected the blood pressure measurements, the SBP, DBP and PP values ​​at the 0thhour in the ready meal group should have been found to be significantly higher than the canned and fresh food group.Urinary BPA concentration showed a statistically significant increase in all three groups at 2 hours after consuming the intervention meals compared to 0 hour measurements. However, there was no statistically significant difference between the groups in blood pressure measurements at the 2nd hour.

With the consumption of intervention meals, there was an increase of 9.682 μg/g in urinary BPA concentrations at 2ndhour in the ready meal group compared to baseline, while there was an increase of 7,.74 μg/g in urinary BPA concentrations in the canned meal group. This was due to the higher BPA concentrations exposed from the meals in the ready meal group. Therefore, while the urinary BPA concentration at the 2nd hour in the ready meal group was significantly higher than the canned and fresh food groups, the SBP and PP values ​​at the 2nd hour were significantly higher than the SBP and PP values ​​at 0th hour in the ready meal group.

Point 5, Question 2: Second, 0th hour BPA, BPS, and BPF samples were collected from Day 1 and Day 2, which are considered “bisphenol washout days” intended to reduce chemical exposure. Could the authors explain why, even after 2 days of this activity, are the baseline values still higher than the 6th hour values after bisphenol intake (and such a difference is statistically significant)?

Response 5, Question 2: “Bisphenol washout days” were only achieved by reducing the dietary sources of BPA. The fact that urinary BPA concentration values ​​at hour 0 were similarly high in all three groups is thought to be due to dermal absorption (personal care products such as thermal paper, paint, shampoo, shower gel, dental materials and cosmetics) and inhalation, which are other exposure sources of BPA.

This topic is mentioned in lines 417-420, “Before the intervention, during a period when multiple dietary sources of bisphenol exposure were controlled, urinary BPA concentrations at 0 h were high in all the three groups. These common exposures are likely associated with direct contact with BPA containing materials or BPA exposure in household dust or indoor air [35].

Therefore, with the consumption of intervention meals, the urinary BPA concentration, which peaked at the 2nd hour in the study, showed a statistically significant decrease in the 6-hour period, so the 2nd hour was >0th hour >4th hour>6thhour. These results indicate that urinary BPA concentrations show a clockwise decrease.

In all three groups, BPA increased in the second hour compared to before the intervention, decreased to its pre-intervention level over time, and gradually decreased to its minimum level in the sixth hour, as it has a half-life of 6 hours.In addition, the same result in all three groups shows that the participants in all three selected groups are homogeneous and there are no differences in BPA metabolism between the groups.

Human pharmacokinetics after single exposures were first reported by Völkel and colleagues (Völkel W, Colnot T, Csanády GA, Filser JG, Dekant W. Metabolism and kinetics of bisphenol a in humans at low doses following oral administration. Chem Res Toxicol. 2002 Oct;15(10):1281-7. doi: 10.1021/tx025548t. PMID: 12387626.). In this study, they gave six volunteers deuterium-labeled BPA (5 mg) by mouth and followed elimination in both blood and urine for 42 hr. Urine levels, measured every 6 hr, peaked 6 hr after administration (19.1 μmol; 4,360 ng/mL); the authors reported a urinary elimination half-life of 5.4 hr. Teeguarden and colleagues (Teeguarden JG, Waechter JM Jr, Clewell HJ 3rd, Covington TR, Barton HA. Evaluation of oral and intravenous route pharmacokinetics, plasma protein binding, and uterine tissue dose metrics of bisphenol A: a physiologically based pharmacokinetic approach. Toxicol Sci. 2005 Jun;85(2):823-38. doi: 10.1093/toxsci/kfi135. Epub 2005 Mar 2. PMID: 15746009.) later used these data in the construction of a physiologically based pharmacokinetic model. The peak plasma concentrations for BPA, BPS, and BPF and half-life of BPA were mentioned in the before-submitted manuscript in lines 419-423 as “In all the three groups, urinary BPA, BPS, and BPF concentrations peaked at the second hour and subsequently decreased. Similarly, in a study by Teeguarden et al., the time to peak urinary BPA concentrations was 2.75 h [41], whereas in the study by Peng et al., urinary BPA concentrations peaked at the fourth hour [37]. BPA has a half-life of approximately 6 h, and elimination of orally administered BPA is completed within 24 h [42].

Point 5, Question 3: Third, did the authors compare the metabolism rate of BPA among three different groups? For example, by comparing the concentration curve (time after BPA intake versus BPA concentration) to investigate if these three groups had similar metabolism rate of BPA. In other words, differences in BPA metabolism among the three groups could potentially contribute to the outcomes observed in the study (e.g., blood pressure and heart rate).

Response 5, Question 3: The metabolism rate of bisphenol derivatives was not compared in this study as this was not the aim when we started the project. Several factors should be considered while evaluating the metabolism rate.

Metabolism rate due to Phase I or Phase II biotransformation enzymes can differ among humans. On the other hand, environmental factors, exposure to different chemical/physical/biological agents, age, sex and lifestyle can also affect metabolism. This is an intervention descriptive study in which dietary factors were considered as a particular effect on the bisphenol levels as well as on blood pressure and heart rate. Therefore, only the dietary intervention was considered as the main effector. However, we plan to measure the effect of metabolism (i.e. the genetic polymorphisms, kidney functions, liver functions, etc.) in our future studies.

Point 6: Discrepancy between statement and data.

First, in lines 365 to 366, the authors state, “Intragroup comparisons of participants in the canned meal group showed no statistically significant difference in SBP and HR values at 0, 2, 4, and 6 h (p > 0.05).” However, according to Table 4, there is statistically significant difference in HR (p = 0.032).

Second, in lines 366 to 367, the authors state, “However, statistically significant differences were detected in DBP and PP values at 0, 2, 4, and 6 h (p < 0.05).” However, according to Table 4, there is no statistically significant difference in PP (p = 0.789).

Third, in lines 368 to 369, the authors state, “Upon analyzing these differences, participants in the canned meal group showed significantly lower DBP and PP values at the 4th hour than those at the 6th hour.” However, according to Table 4, the PP value at 6th hour is not statistically different from that at 4th hour.

Response 6: In lines 371-373, the sentence is corrected as “Intragroup comparisons of participants in the canned meal group showed no statistically significant difference in SBP and PP values at 0, 2, 4, and 6 h (p > 0.05). However, statistically significant differences were detected in DBP and HR values at 0, 2, 4, and 6 h (p < 0.05). Upon analyzing these differences, participants in the canned meal group showed significantly lower DBP and HR values at the 4th hour than those at the 6th hour.

Point 7: Typographical errors.

First, in line 236, “and washed and washed”.

Second, in line 217, the unit “ng/mL” should come after 0.84. In line 248, the unit “ng/mL” should come after 0.94.

Third, in the footnote of Table 1, the “*” is missing from “p < 0.05”.

Forth, in both Tables 3 and 4, “a, b, and c” were used to indicate “canned meal group, ready-to-eat group, and fresh meal group”. Meanwhile, “a, b, c, and d” were also used to indicate “0, 2, 4, 6 hour”. This could potentially cause confusion for the audience. The authors might consider using different symbols (e.g., I, II, III, IV) to represent the hours instead.

Response 7: In line 239, the expression of “and washed” was deleted and in lines 237-240, the expression is changes to “The stainless steel stirrer was thoroughly cleaned (using detergents and solvents) after homogenization of each food sample to prevent cross-contamination between samples and washed with n-hexane:tetrahydrofuran (1:1, v/v) and dried in an in-cubator for 4 h.”

In line 219-221, the sentence is corrected to “The limits of detection were 0.21, 0.88 and 0.84 ng/mL for BPA, BPF, and BPS, respectively.”

In lines 251-252, the sentence is corrected to “The limits of detection for food samples were 0.52, 0.99 and 0.94 ng/mL for BPA, BPF, and BPS, respectively.”

In the footnote of Table 1, "*" was added to "p < 0.05".

In Table 3 and Table 4, different symbols (I, II, III, IV) were used to indicate "0th, 2nd, 4th, 6th hours".

We would like to thank Reviewer 1 for his/her suggestions and contributions to the text.

Reviewer 2 Report

Comments and Suggestions for Authors

Lines 188-189: " The systolic BP (SBP), diastolic BP (DBP), and HR of the participants were measured using a sphygmomanometer before and 2, 4, and 6 h after the diet intervention." What kind of sphygmomanometer?

Line 192: "The averages of two SBP and DBP measurements and two HR measurements were used  for statistical analysis" From the theoretical point of view, drawing a mean from two measurements is difficult to accept in mathematical terms while not protecting from potential biases linked to the inherent variability of BP

Lines 301-304: "..the mean urinary BPA concentration of participants in the ready-to-eat meal group at 0 and 2 h (12.615 ± 3.725 and 22.297 ± 6.368 μg/g creatinine,) was significantly higher than those in the canned and fresh meal groups  (9.149 ± 3.904, 16.323 ± 7.353 and 7.258 ± 1.314, 14.319 ± 3.268 μg/g creatinine, respectively)" Do you believe that this difference in BPA concentration is pathophysiologically relevant?

 Lines 357-359: " Intergroup comparisons showed no statistically significant difference in HR and PP values of participants in the canned, ready-to-eat, and fresh meal groups at 0, 2, 4, and 6 h (p > 0.05)." and Lines 360-363: "... Similarly, no significant difference was observed in SBP and DBP values at 0, 2, and 4 h (p>0.05). However, at the sixth hour, a statistically significant difference was identified in SBP and DBP values among participants in the canned, ready-to-eat, and fresh meal groups (p<0.05). Upon examining this difference, participants in the canned meal group showed significantly higher SBP and DBP values than those in the ready-to-eat meal group." Changes in SBP and DBP increase may bear some statistical significance but have in my opinion, no pathophysiological and even less clinical meaning.

Lines 470-471: "These results demonstrate the acute effects of increased BPA, BPS, and BPF exposure due to ready-to-eat food consumption on SBP and HR." These results at best suggest some cardiovascular effect of BPA and, if ever, do not support the impact of BPS and BPF at least if I understood correctly what lines 432-433 report "No significant differences were noted in urinary BPS and BPF concentrations among participants in the canned, ready-to-eat, and fresh meal groups following meal consumption."

Lines 474-475: "Furthermore, a decrease in PP is generally an indicative of reduced cardiac function [43]." Wrong conclusion when applied to a cohort of young males and females.

Lines 495-498: " Therefore, the significant increase in SBP and DBP values at the 6th hour in the canned meal group compared with that in the ready-to-eat meal group may be attributed to the synergistic effect of BPA, BPS, and BPF exposure." Highly speculative conclusion unsupported by the available data

In conclusion, I have no objection with the experimental design of the study which I find is by and large negative and in need of a more detailed and balanced discussion.

Author Response

Response to Reviewer 2

Thank you very much for taking the time to review this manuscript. Please find the detailed responses below and the corresponding revisions/corrections highlighted changes in the re-submitted files.

Point 1: Lines 188-189: "The systolic BP (SBP), diastolic BP (DBP), and HR of the participants were measured using a sphygmomanometer before and 2, 4, and 6 h after the diet intervention." What kind of sphygmomanometer?

Response 1: Omron brand M3 Comfort (HEM-7155-E(C)) automatic blood pressure monitor was used. This information is now added to the text.

Point 2: Line 192: "The averages of two SBP and DBP measurements and two HR measurements were used for statistical analysis" From the theoretical point of view, drawing a mean from two measurements is difficult to accept in mathematical terms while not protecting from potential biases linked to the inherent variability of BP. 

Response 2: In this study, the relevant publication of the American Heart Association (AHA) (Recommendations for Blood Pressure Measurement in Humans and Experimental Animals. 2005;45(1):142–61) was taken into consideration when blood pressure measurements were performed. The AHA used the following expression for blood pressure measurement in the manuscript; “A minimum of 2 readings should be taken at intervals of at least 1 minute, and the average of those readings should be used to represent the patient’s blood pressure. If there is >5 mm Hg difference between the first and second readings, additional (1 or 2) readings should be obtained, and then the average of these multiple readings is used.” Therefore, according to the AHA guidelines, the average of the two measurements represents blood pressure.

Point 3: Lines 301-304: ".the mean urinary BPA concentration of participants in the ready-to-eat meal group at 0 and 2 h (12.615 ± 3.725 and 22.297 ± 6.368 μg/g creatinine,) was significantly higher than those in the canned and fresh meal groups  (9.149 ± 3.904, 16.323 ± 7.353 and 7.258 ± 1.314, 14.319 ± 3.268 μg/g creatinine, respectively)" Do you believe that this difference in BPA concentration is pathophysiologically relevant?

Response 3: BPA is recognized as an endocrine disrupting chemical by many leading organizations, such as US EPA, EFSA, WHO. In EFSA and FAO/WHO and the toxicological effects of BPA on health are widely present in literature. Moreover, the tolerable daily intake amount has been determined. In the literature, it has been proven that even long-term low dose exposure to bisphenols has a pathophysiological effect (i.e. “low dose effect”). Therefore, in the light of scientific evidence, statistically significant differences in BPA concentrations are predicted to produce significant effects. Although it is hard to particularly indicate the pathophysiological effects of bisphenol derivatives as humans are exposed to a wide range of endocrine disruptors as a mixture in daily life, animal studies as well as case control studies suggest small differences in biological fluid levels might be the cause of several endocrine system disorders as also given in our different studies (Durmaz E, Asci A, Erkekoglu P, Balcı A, Bircan I, Koçer-Gumusel B. Urinary bisphenol A levels in Turkish girls with premature thelarche. Hum Exp Toxicol. 2018 Oct;37(10):1007-1016. doi: 10.1177/0960327118756720. Epub 2018 Feb 6. PMID: 29405766.; Durmaz E, AÅŸçı A, ErkekoÄŸlu P, Akçurin S, GümüÅŸel BK, Bircan I. Urinary bisphenol a levels in girls with idiopathic central precocious puberty. J Clin Res Pediatr Endocrinol. 2014;6(1):16-21. doi: 10.4274/Jcrpe.1220. PMID: 24637305; PMCID: PMC3986734.; Akgül S, Sur Ü, Düzçeker Y, Balcı A, Kızılkan MP, Kanbur N, BozdaÄŸ G, ErkekoÄŸlu P, GümüÅŸ E, Kocer-Gumusel B, Derman O. Bisphenol A and phthalate levels in adolescents with polycystic ovary syndrome. Gynecol Endocrinol. 2019 Dec;35(12):1084-1087. doi: 10.1080/09513590.2019.1630608. Epub 2019 Jun 20. PMID: 31219355.; Sunman B, Yurdakök K, Kocer-Gumusel B, Özyüncü Ö, Akbıyık F, Balcı A, Özkemahlı G, ErkekoÄŸlu P, Yurdakök M. Prenatal bisphenol a and phthalate exposure are risk factors for male reproductive system development and cord blood sex hormone levels. Reprod Toxicol. 2019 Aug;87:146-155. doi: 10.1016/j.reprotox.2019.05.065. Epub 2019 Jun 3. PMID: 31170452.; Kondolot M, Ozmert EN, Ascı A, Erkekoglu P, Oztop DB, Gumus H, Kocer-Gumusel B, Yurdakok K. Plasma phthalate and bisphenol a levels and oxidant-antioxidant status in autistic children. Environ Toxicol Pharmacol. 2016 Apr;43:149-58. doi: 10.1016/j.etap.2016.03.006. Epub 2016 Mar 9. PMID: 26991849.). 

Point 4: Lines 357-359: " Intergroup comparisons showed no statistically significant difference in HR and PP values of participants in the canned, ready-to-eat, and fresh meal groups at 0, 2, 4, and 6 h (p > 0.05)." and Lines 360-363: "... Similarly, no significant difference was observed in SBP and DBP values at 0, 2, and 4 h (p>0.05). However, at the sixth hour, a statistically significant difference was identified in SBP and DBP values among participants in the canned, ready-to-eat, and fresh meal groups (p<0.05). Upon examining this difference, participants in the canned meal group showed significantly higher SBP and DBP values than those in the ready-to-eat meal group." Changes in SBP and DBP increase may bear some statistical significance but have in my opinion, no pathophysiological and even less clinical meaning.

Response 4: This study is a descriptive toxicological study and does not present pathophysiological results. The statistically significant results were found in blood pressure levels in healthy individuals in the study and this indicates that bisphenol exposure may cause an increase in the likelihood of cardiovascular disease in healthy individuals in the future. However, other risk factors such as genetic susceptibility, environmental exposures, lifestyle, cigarette/alcohol consumption, obesity/metabolic syndrome, presence of another disease (i.e. cardiac diseases, diabetes) can also affect the outcome. Therefore, being genetically susceptible as well as exposures can act synergistically with bisphenol exposure to have hypertension and related pathological conditions.

Point 5: Lines 470-471: "These results demonstrate the acute effects of increased BPA, BPS, and BPF exposure due to ready-to-eat food consumption on SBP and HR." These results at best suggest some cardiovascular effect of BPA and, if ever, do not support the impact of BPS and BPF at least if I understood correctly what lines 432-433 report "No significant differences were noted in urinary BPS and BPF concentrations among participants in the canned, ready-to-eat, and fresh meal groups following meal consumption."

Response 5: In the ready-to-eat meal group, urinary BPA, BPS and BPF concentrations in the 2nd hour after consumption of the intervention meal were found to be statistically significantly increased compared to the 0th hour. Therefore, it refers to the acute effect of increased BPA, BPS, and BPF exposure on blood pressure measurements, not only BPA exposure.

Point 6: Lines 474-475: "Furthermore, a decrease in PP is generally indicative of reduced cardiac function [43]." Wrong conclusion when applied to a cohort of young males and females.

Response 6: In the study, the reason for the decrease in PP value in healthy young people is explained in lines 477-478 as “The increase in PP may be a consequence of the increase in SBP and the decrease in DBP.”. In addition, the sentence in lines 478-479 is a descriptive statement used by referring to literature on PP.

Point 7: Lines 495-498: " Therefore, the significant increase in SBP and DBP values at the 6th hour in the canned meal group compared with that in the ready-to-eat meal group may be attributed to the synergistic effect of BPA, BPS, and BPF exposure." Highly speculative conclusion unsupported by the available data

Response 7: In order not to cause any speculation as the reviewer has suggested the findings and conclusions in lines 504-506 are explained and supported by referring to the literature in lines 502-505 as “A previous study also suggested that simultaneous exposure to bisphenol analogs along with low BPA concentrations lead to more severe cardiovascular or cardiometabolic disorders or toxicity than exposure to high BPA concentrations alone because of the synergistic effect [46].”

Point 8: In conclusion, I have no objection with the experimental design of the study which I find is by and large negative and in need of a more detailed and balanced discussion.

Response 8: The reviewer is right that such descriptive studies does not clearly demonstrate a “cause-effect relationship”. However, as it can be seen from our results, exposure to bisphenols can lead to acute changes in blood pressure. We need more mechanistic studies in order to clearly show a cause-effect association. More in vitro as well as in vivo studies are needed to show the mechanism of action of bisphenols on heart, veins, plaque formation, endothelial cells and vascular tension/pressure. Our study only suggests there might be relation with the orally taken bisphenols and heart rate/blood pressure. This is an intervention study and it was hard for us to persuade the subjects to consume the types of diets we used. We agree that more mechanistical studies are needed. We believe that descriptive studies pave the way for mechanistic studies and that the mechanism of the relationship between heart rate and blood pressure and bisphenol exposure will be revealed with new mechanistic studies in the future.

We would like to thank Reviewer 2 for his/her suggestions and contributions to the text.

Round 2

Reviewer 1 Report

Comments and Suggestions for Authors

Thank you to the authors for addressing my comments. I have one minor suggestion for the revised manuscript. Could the authors include a description of how the study meal was selected? As mentioned in my previous comment, total BPA concentrations can vary depending on the food consumed. Providing the rationale behind the food selection would be helpful for understanding the study design.

Reviewer 2 Report

Comments and Suggestions for Authors

This Reviewer cannot but remark that the specific questions raised about this document were as a matter of fact unanswered. 
